# Associations between specialty care and improved outcomes among patients with diabetic foot ulcers

Yingzhou Liu[1☯], Menggang Yu[1☯], Jamie N. LaMantia[2], Jennifer Mason Lobo[3], Justin J. Boutilier[4], Yao Liu[5], Meghan B. Brennan[2]*

1 Department of Biostatistics and Medical Informatics, School of Medicine and Public Health, University of Wisconsin, Madison, Wisconsin, United States of America, 2 Department of Medicine, School of Medicine and Public Health, University of Wisconsin, Madison, Wisconsin, United States of America, 3 Department of Public Health Sciences, University of Virginia, Charlottesville, Virginia, United States of America, 4 Department of Industrial and Systems Engineering, College of Engineering, University of Wisconsin, Madison, Wisconsin, United States of America, 5 Department of Ophthalmology and Visual Sciences, School of Medicine and Public Health, University of Wisconsin, Madison, Wisconsin, United States of America

☯ These authors contributed equally to this work.
* mbbrennan@medicine.wisc.edu

## Abstract

### Objective

Specialty care may improve diabetic foot ulcer outcomes. Medically underserved populations receive less specialty care. We aimed to determine the association between specialty care and ulcer progression, major amputation, or death. If a beneficial association is found, increasing access to specialty care might help advance health equity.

### Research design and methods

We retrospectively analyzed a cohort of Wisconsin and Illinois Medicare patients with diabetic foot ulcers (n = 55,409), stratified by ulcer severity (i.e., early stage, osteomyelitis, or gangrene). Within each stratum, we constructed Kaplan-Meier curves for event-free survival, defining events as: ulcer progression, major amputation, or death. Patients were grouped based on whether they received specialty care from at least one of six disciplines: endocrinology, infectious disease, orthopedic surgery, plastic surgery, podiatry, and vascular surgery. Multivariate Cox proportional hazard models estimated the association between specialty care and event-free survival, adjusting for sociodemographic factors and comorbidities, and stratifying on ulcer severity.

### Results

Patients who received specialty care had longer event-free survival compared to those who did not (log-rank p<0.001 for all ulcer severity strata). After adjusting, receipt of specialty care, compared to never, remained associated with improved outcomes for all ulcer severities (early stage adjusted hazard ratio 0.34, 95% CI 0.33–0.35, p<0.001; osteomyelitis aHR 0.22, 95% CI 0.20–0.23, p<0.001; gangrene aHR 0.22, 95% CI 0.20–0.24, p<0.001).

**Data Availability Statement:** Data that support the findings of this study are available from Centers for Medicare and Medicaid Service and the American

Medical Association, but require a data use agreement and are not publicly available. Readers can contact Lauren Gee at lgee@medicine.wisc.edu, Senior Research Administrator for the University of Wisconsin Department of Medicine, for any data access requests.

**Funding:** This work was supported by the following grants: Agency for Healthcare Research and Quality K08 HS026279-01A1 (MBB; ahrq.gov) and National Institute for Diabetes and Digestive and Kidney Diseases R01 DK132569-01 (MBB; niddk.nih.gov). Additional funding came from an institutional grant from Research to Prevent Blindness, Inc. to the University of Wisconsin's Department of Ophthalmology and Visual Sciences (YL; rpbusa.org). The funders had no role in study design, data collection and analysis, decision to publish, or preparation of the manuscript.

**Competing interests:** The authors have declared that no competing interests exist.

## Conclusions

Specialty care was associated with longer event-free survivals for patients with diabetic foot ulcers. Increased, equitable access to specialty care might improve diabetic foot ulcer outcomes and disparities.

## Introduction

Nearly 25% of patients with diabetes will develop a foot ulcer during their lifetime [1]. Of those, over half will die or undergo major (i.e., above-ankle) amputation within five years [2]. Most American patients fear major amputation more than death [3]. There is an urgent need to reduce the morbidity and mortality associated with amputation due to its rising incidence and the disproportionate impact on medically underserved populations [4,5].

Poor outcomes and mounting disparities are the result, in part, of failing to apply what we know works [6]. Data from single center studies demonstrate that multidisciplinary, specialty teams are associated with reductions in major amputations [7]. Their success may be due to consistently applying proven limb salvage strategies. However, we do not know if specialty care in general is associated with better outcomes [7]. Currently, patients may be treated entirely by primary care providers or hospitalists. Others may be referred to specialty care spanning a variety of different disciplines and varying considerably based, in part, on local workflows and regional availablilty. If specialty care is associated with improved outcomes, increasing access to specialists might be particularly impactful for medically underserved patients, who are less likely to see a specialist for a foot ulcer and are at higher risk of amputation [8,9].

To date, studies investigating specialty care outside of multidisciplinary teams have been restricted mainly to podiatry, vary based on stage of care (i.e., prevention vs. treatment), and demonstrate mixed results [10]. To address these gaps, we conducted a multi-state, retrospective cohort study of specialty care for patients with active diabetic foot ulcers. Specialty care was defined as receiving care from at least one of the six most common disciplines represented on multidisciplinary teams [7]. We aimed to determine whether specialty care—regardless of whether it is part of an organized, multidisciplinary effort—is associated with longer times from ulcer diagnosis to progression, amputation, or death. We hypothesized that patients who received specialty care would have longer event-free survivals.

## Methods

### Data sources

We evaluated a 100% cohort of Medicare fee-for-service beneficiaries residing in Wisconsin (2009–2017) and Illinois (2012–2017) using Part A and B claims data for ambulatory and hospitalized care, obtained through the Centers for Medicare and Medicaid Services Chronic Condition Data Warehouse. We supplemented Medicare provider taxonomy codes with American Medical Association Masterfile datasets to better identify the following specialties, whose clinicians often register with Medicare prior to finishing specialty training: endocrinology, infectious disease, and vascular surgery [11]. The authors worked with a deidentified dataset first accessed on April 4, 2022.

### Study design

We built a retrospective, rolling cohort of Medicare beneficiaries ages 21 through 89 years (inclusive) and diagnosed with incident diabetic foot ulcers between January 1, 2009 and

December 31, 2017. We included a 2-year baseline period with continuous Medicare coverage to identify those with diabetes and incident foot ulcers using validated claims algorithms able to distinguish ulcer severity (S1 Table) [12,13]. Exclusion criteria were as follows: presence of a major amputation or foot ulcer during the baseline period, or railroad benefits. Patients were followed from the date of diabetic foot ulcer diagnosis until major amputation, death, loss of Medicare part A and B coverage, or the study end date (December 31, 2017).

We constructed three strata based on ulcer severity (i.e., early stage, osteomyelitis, or gangrene) to account for confounding by indication, meaning that patients with more advanced ulcers would be more likely to both receive specialty care and have poor outcomes (Fig 1, S1 Table) [12]. Patients moved to a more advanced stratum on the date that the ulcer was diagnosed to have progressed. Once patients were classified with a more advanced ulcer, they were not allowed to re-enter a less severe ulcer stratum. Due to the de-identified nature of the dataset, informed consent was waived and this study was granted an exemption by the University of Wisconsin Health Sciences Institutional Review Board (2019–0442).

## Outcomes

We collected three types of variables for this study, all of which are listed in Table 1: outcome variables, primary explanatory variables, and covariates. Our primary, composite outcome was event-free survival, where an event was defined as either progression to a more advanced ulcer stratum, major amputation, or death. Event-free survival was measured in days from entry into an ulcer stratum (i.e., early stage, osteomyelitis, or gangrene) to one of these three events. Major amputation was identified using *International Classification of Diseases*, $9^{th}$ *Revision* (ICD-9) or current procedural terminology codes [14]. Our secondary outcome was amputation-free survival, censoring for disease progression.

## Primary explanatory variable

We defined specialty care as receiving care from at least one of the following six disciplines, representing the most common specialties on multidisciplinary teams caring for patients with diabetic foot ulcers: endocrinology, infectious disease, orthopedic surgery, plastic surgery, podiatry, and vascular surgery [7]. We included specialists identified as such by either Medicare taxonomy codes or supplemental American Medical Association Masterfile data to

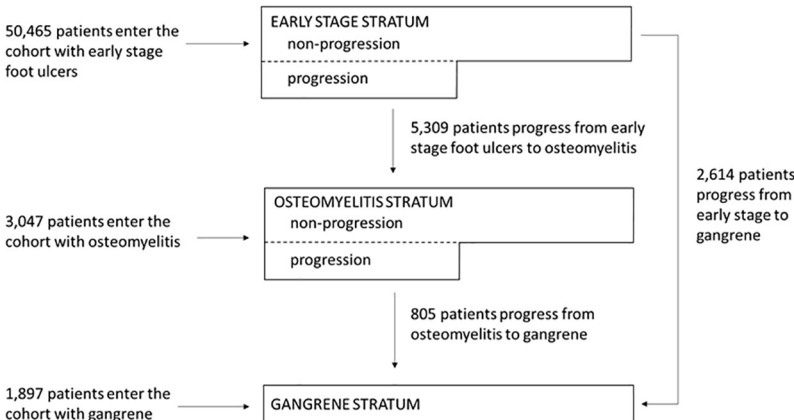

**Fig 1. Flow diagram of patient entry into and progression from the three ulcer severity strata.**

**Table 1. Patient characteristics at cohort entry.**

| Characteristic | Total cohort n = 55,409 | Patients who did not receive specialty care n = 15,942 (28.77) | Patients who received care from at least 1 specialist n = 39,467 (71.23) |
|---|---|---|---|
| **Sociodemographics** | | | |
| Age in years, mean (SD) | 72.98 (10.70) | 73.75 (10.85) | 72.67 (10.63) |
| Male | 29,441 (53.13) | 8,799 (55.19) | 20,642 (52.30) |
| Race & ethnicity | | | |
| Black | 7,284 (13.15) | 2,383 (14.95) | 4,901 (12.42) |
| Non-Hispanic white | 45,639 (82.37) | 12,834 (80.50) | 32,805 (83.12) |
| Other/unknown* | 2,486 (4.49) | 725 (4.55) | 1,761 (4.46) |
| Rurality† | | | |
| Urban | 45,964 (82.95) | 12,808 (80.34) | 33,156 (84.01) |
| Rural | 9,415 (16.99) | 3,124 (19.60) | 6,291 (15.94) |
| Medicaid | 13,832 (24.96) | 4,721 (29.61) | 9,111 (23.09) |
| **Comorbidities and Ulcer Severity** | | | |
| Comorbidities | | | |
| History of myocardial infarction | 14,731 (26.59) | 4,873 (30.57) | 9,858 (24.98) |
| History of heart disease | 44,356 (80.05) | 13,139 (82.42) | 31,217 (79.10) |
| History of stroke | 16,885 (30.47) | 5,662 (35.52) | 11,223 (28.44) |
| History of eye disease | 16,539 (29.85) | 4,200 (26.35) | 12,339 (31.26) |
| History of peripheral vascular disease | 46,596 (84.09) | 13,582 (85.20) | 33,014 (83.65) |
| History of renal disease | 20,450 (36.91) | 7,449 (46.73) | 13,001 (32.94) |
| Ulcer severity at cohort entry | | | |
| Early | 50,465 (91.08) | 13,580 (85.18) | 36,885 (93.46) |
| Osteomyelitis | 3,047 (5.50) | 1,174 (7.36) | 1,873 (4.75) |
| Gangrene | 1,897 (3.42) | 1,188 (7.45) | 709 (1.80) |
| **Specialty Care** | | | |
| Endocrinology | 6,409 (11.57) | — | 6,409 (16.24) |
| Infectious Disease | 5,404 (9.75) | — | 5,404 (13.69) |
| Orthopedic surgery | 11,350 (20.48) | — | 11,350 (28.76) |
| Plastic surgery | 1,516 (2.74) | — | 1,516 (3.84) |
| Podiatry | 32,952 (59.47) | — | 32,952 (83.49) |
| Vascular Surgery | 5,683 (10.26) | — | 5,683 (14.40) |
| Mean number of specialists seen (range) | 1.14 (0–6) | — | 1.60 (1–6) |
| **Reason for Censoring** | | | |
| End of the study period | 27,760 (50.10) | 5,231 (32.81) | 22,529 (57.08) |
| Death | 14,242 (25.70) | 5,926 (37.17) | 8,316 (21.07) |
| Major amputation | 7,089 (12.79) | 3,081 (19.33) | 4,008 (10.16) |
| Loss of Medicare coverage/Lost to follow-up | 6,318 (11.40) | 1,704 (10.69) | 4,614 (11.69) |

Data are presented as n (%) unless otherwise noted.

* Includes 377 patients identifying as American Indian/Alaskan Native, 409 patients identifying as Asian/Pacific Islander, 1441 patients identifying as other races or ethnicities, and 259 patients whose race/ethnicity is unknown.

† Columnar percents for rurality do not total 100% due to missing values, which were suppressed due to low count numbers in accordance with Medicare standards.

maximize sensitivity [11]. We classified patients as having received specialty care once they had any specialty visit that coded for the ulcer, except for endocrinologists who were allowed to code for diabetes rather than the ulcer itself following ulcer diagnosis. We excluded vascular or orthopedic care 2 months prior to major amputation due to concerns that the reason for referral to these surgeons was solely for major amputation rather than limb salvage. The 2-month cut-off was based on clinical experience and histogram plots of surgical care timing relative to major amputation (S1 Fig). Specialty care was modeled as a categorical variable relative to patients entering an ulcer stratum: never (reference), before, or after entry. The "before" category was only created for the osteomyelitis and gangrene strata and was composed of patients who entered the study with earlier stage ulcers, received specialty care at that point, and then entered the osteomyelitis or gangrene strata when their ulcers progressed.

## Covariates

We included the following sociodemographics, assessed at study entry: age, sex, race and ethnicity, rurality, and whether the patient had received Medicaid. Rurality was defined using consolidated Rural Urban Commuting Area (RUCA) codes: urban (RUCA 1–3) or rural (RUCA 4–10) [15]. We identified the following comorbidities using validated claims algorithms: eye disease, heart disease, myocardial infarction, peripheral vascular disease, renal disease, and stroke [16,17].

## Statistical analysis

We described patient sociodemographics, comorbidities, and ulcer severity at the time of cohort entry, the type of specialty care provided when applicable, and reason for censoring both overall and stratified by whether the patient had received specialty care during the study period. We constructed Kaplan-Meier survival curves, stratified by ulcer severity, to compare marginal difference in our primary and secondary outcomes using the log rank test. Within each stratum, patients were grouped based on whether they received specialty care: never, before (for osteomyelitis and gangrene strata only), or after stratum entry. We built a multivariable Cox proportional hazard model, stratified by ulcer severity, to characterize the relationship between specialty care and our primary outcome. We used stepwise covariate selection to build the most parsimonious model using the My.stepwise R software package [18]. We set entry and stay criteria for our final model to p-values ≥0.20. This same approach was used to model our secondary outcome. We assessed possible residual confounding by calculating risk scores for each individual in the cohort [19]. Risk scores were generated with a multivariate Cox model fit to the group who never received specialty care. We used Wilcoxon rank sum tests to describe differences in time to ulcer progression based on whether patients received specialty care when they had an earlier stage ulcer or more advanced ulcer. To account for possible confounding in receiving specialty care, robust propensity score-based analyses were also conducted using the CBPS R package.

## Data and resource availability

Data that support the findings of this study are available from Centers for Medicare and Medicaid Service and the American Medical Association, but require a data use agreement and are not publicly available. Data are, however, available from the authors upon reasonable request and with permission of both entities.

## Results

A total of 55,409 patients with diabetic foot ulcers were included. Over 90% entered the cohort with an early stage ulcer (Fig 1, Table 1). The average length of follow-up was 20.14 months. During the study, 5,309 (11%) patients initially diagnosed with early stage ulcers progressed to osteomyelitis. Of those initially diagnosed with early stage ulcers or osteomyelitis, 3,419 (6%) patients developed gangrene. Within the total cohort, 7,089 (13%) patients underwent a major amputation and 14,242 (26%) patients died during follow-up. The rate of major amputation or death was higher within the osteomyelitis and gangrene strata, compared to the early stage stratum (S2 Fig).

Over 71% (n = 39,467) of the total cohort saw at least one specialist during their disease course, most commonly podiatry. Among those who received specialty care, the mean number of disciplines involved was 1.60 (range: 1–6; Table 1). Compared to those who never saw a specialist, a higher proportion of patients who received specialty care lived in urban areas and identified as non-Hispanic White. A lower proportion of patients who received specialty care were covered by Medicaid or state safety-net insurance. Comorbidities were relatively evenly distributed between those who did and did not receive specialty care. A larger proportion of patients who received specialty care had ulcers complicated by osteomyelitis or gangrene (Table 1).

Within the early stage stratum, patients who saw a specialist had longer event-free survivals compared to those who did not (log-rank p<0.001; Fig 2). Specifically, half the patients who saw a specialist met the primary composite endpoint by day 1502, compared to day 249 for those who did not see a specialist. Patients who received specialty care also had longer amputation-free survivals (log-rank p<0.001; S2 Fig).

In the adjusted Cox proportional hazard model of the early stage stratum, receiving specialty care was associated with a significant reduction in the hazard of the primary composite endpoint compared to not receiving specialty care (aHR 0.34, 95% CI 0.33–0.35, p<0.001; Table 2). Receiving specialty care was also associated with a reduced hazard for major amputation or death compared to those who did not receive specialty care (aHR 0.29, 95% CI 0.28–0.30, p<0.001; Table 2).

Within the osteomyelitis stratum, patients who saw a specialist had longer event-free survivals compared to those who were not seen by a specialist (log-rank p<0.001; Fig 2). Half the patients with osteomyelitis who never saw a specialist met the primary composite endpoint by day 7. Half of those that saw a specialist for the first time *before* they were diagnosed with osteomyelitis met the primary composite endpoint by day 115 compared to day 1149 among those who saw a specialist for the first time *after* diagnosis of osteomyelitis. We observed similar findings for amputation-free survival (S2 Fig).

In the adjusted Cox proportional hazard model of the osteomyelitis stratum, receiving specialty care was associated with a significant reduction in the hazard of the primary composite endpoint compared to patients who did not receive specialty care (Table 2). We observed this associated reduction for both patients who received specialty care before and after the diagnosis of osteomyelitis. However, the estimated reduction was greatest for those receiving specialty care after the diagnosis of osteomyelitis (aHR$_{after}$ 0.22, 95% CI 0.20–0.23, p<0.001; aHR$_{before}$ 0.49, 95% CI 0.46–0.52, p<0.001). These trends were similar to those for amputation-free survival (Table 2).

Outcomes within the gangrene stratum were generally poor, although patients who received specialty care continued to have longer event-free survival (log rank p<0.001; Fig 2). Patients who had their first specialty visit before being diagnosed with gangrene initially had better outcomes than those who never saw a specialist, although the survival curves for both

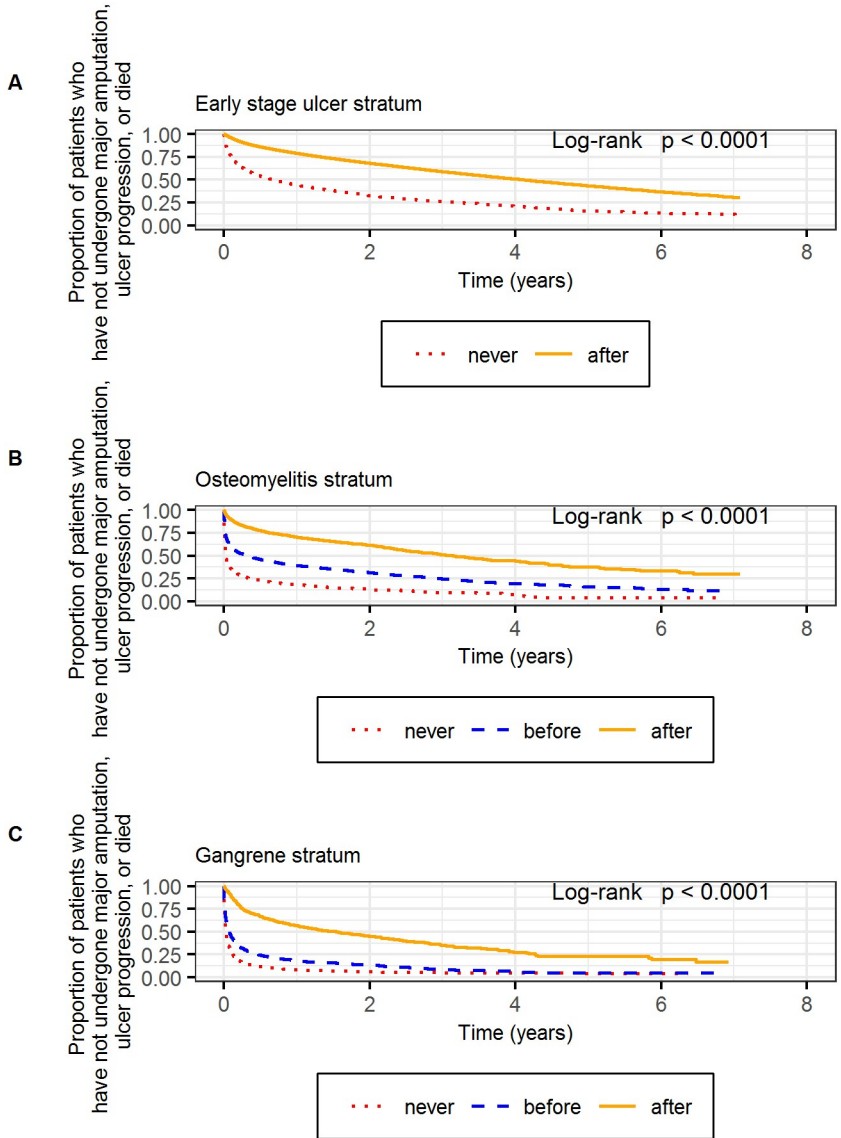

**Fig 2. Kaplan-Meier curves depicting the proportion of patients who have not undergone major amputation, ulcer progression, or death.** Results were stratified based on ulcer severity: early stage (A), osteomyelitis (B), and gangrene (C). Within each stratum, patients were characterized as having never received specialty care, having received specialty care before the diagnosis of osteomyelitis or gangrene (which applied to some of the patients who entered the cohort with a less severe ulcer and progressed to osteomyelitis or gangrene), after entering the stratum (i.e., being diagnosed with an early stage ulcer, osteomyelitis, or gangrene, respectively).

outcomes converged by year 4. Half the patients with gangrene who never saw a specialist had either undergone a major amputation or died by day 9. Half the patients who had their first specialty visit _before_ being diagnosed with gangrene had either undergone a major amputation or died by day 26, compared to day 570 for patients seeing a specialist for the first time _after_ the diagnosis of gangrene. Because our most advanced ulcer categorization was gangrene, our analysis of primary and secondary outcomes were identical for patients in this stratum.

In the adjusted Cox proportional hazard model of the gangrene stratum, receiving specialty care remained associated with a reduced hazard of undergoing major amputation or dying

**Table 2. Adjusted hazard ratios for the relationship between specialty care and the primary outcome of ulcer progression, major amputation or death, & the secondary outcome of major amputation or death, censoring for ulcer progression.**

| Ulcer Severity | Specialty Care | | |
|---|---|---|---|
| | Never | Before entry into stratum | After entry into stratum |
| **Early Stage Stratum** | | | |
| Persons (%) | 14,488 (28.72) | N/A | 35,950 (71.28) |
| Person-years | 11,150 | – | 64,311 |
| aHR for primary outcome (ulcer progression, major amputation, or death) * | 1 (ref) | N/A | 0.34 |
| 95% CI for primary outcome | – | – | 0.33–0.35 |
| aHR for secondary outcome (major amputation or death)† | 1 (ref) | N/A | 0.29 |
| 95% CI for secondary outcome | – | – | 0.28–0.30 |
| **Osteomyelitis Stratum** | | | |
| Persons (%) | 2,255 (27.01) | 3,632 (43.50) | 2,463 (29.50) |
| Person-years | 733 | 2,938 | 3,837 |
| aHR for primary outcome (ulcer progression, major amputation, or death)‡ | 1 (ref) | 0.49 | 0.22 |
| 95% CI for primary outcome | – | 0.46–0.52 | 0.20–0.23 |
| aHR for secondary outcome (major amputation or death)§ | 1 (ref) | 0.49 | 0.22 |
| 95% CI for secondary outcome | – | 0.45–0.52 | 0.20–0.23 |
| **Gangrene Stratum** | | | |
| Persons (%) | 2,242 (42.21) | 2,036 (38.33) | 1,034 (19.47) |
| Person-years | 428 | 843 | 1,287 |
| aHR for primary outcome (ulcer progression, major amputation, or death)‖ | 1 (ref) | 0.62 | 0.22 |
| 95% CI for primary outcome | – | 0.58–0.66 | 0.20–0.24 |
| aHR for secondary outcome (major amputation or death)¶ | 1 (ref) | 0.62 | 0.22 |
| 95% CI for secondary outcome | – | 0.58–0.66 | 0.20–0.24 |

* The Cox proportional hazard model controls for age, sex, race/ethnicity, rurality, heart disease, myocardial infarction, peripheral vascular disease, renal disease, eye disease and stroke.

† The Cox proportional hazard model controls for age, sex, race/ethnicity, rurality, heart disease, myocardial infarction, peripheral vascular disease, renal disease, uncomplicated diabetes and stroke.

‡ The Cox proportional hazard model controls for age, sex, race/ethnicity, rurality, eye disease, heart disease, myocardial infarction, peripheral vascular disease, renal disease, stroke and uncomplicated diabetes.

§ The Cox proportional hazard model controls for age, sex, race/ethnicity, rurality, eye disease, heart disease, myocardial infarction, peripheral vascular disease, renal disease, stroke and uncomplicated diabetes.

‖ The Cox proportional hazard model controls for age, sex, race/ethnicity, rurality, eye disease, myocardial infarction, peripheral vascular disease, renal disease, and uncomplicated diabetes.

¶ The Cox proportional hazard model controls for age, sex, race/ethnicity, rurality, eye disease, myocardial infarction, peripheral vascular disease, renal disease, and uncomplicated diabetes.

compared to patients who did not receive specialty care (Table 2). Similar to the osteomyelitis stratum, patients who received specialty care both before and after gangrene diagnosis had reduced hazards; however, the magnitude of the reduction was greatest for those who received specialty care after diagnosis of gangrene (aHR$_{after}$ 0.22, 95% CI 0.20–0.24, p<0.001; aHR$_{before}$ 0.62, 95% CI 0.58–0.66, p<0.001).

We performed additional, descriptive analysis to explore potential reasons why patients who received specialty care *before* their ulcers progressed had shorter event-free survival after entering the more severe stratum compared to those who received specialty care for the first time *after* ulcer progression. Regarding disease trajectory, patients who received specialty care before ulcer progression spent a longer time in less severe ulcer strata than patients who did not receive specialty care in earlier strata (Wilcoxon rank sum test, p<0.001; S3 Fig). For instance, among those who entered the cohort with early stage ulcers but then had wounds that progressed to osteomyelitis, patients who were seen by specialists when they had early stage ulcers spent a median of 147 days in the early stage stratum before developing osteomyelitis. This is in comparison to 15 days for those who progressed from early stage ulcers to osteomyelitis without seeing a specialist while they had an early stage ulcer (Wilcoxon rank sum test: p<0.001). Regarding timing of specialty care, among those who saw a specialist before their ulcers advanced, specialty care was typically early in the disease course. For instance, among those who entered the cohort with early stage ulcers, received specialty care, and then developed osteomyelitis, the median time between initial specialty care and development of osteomyelitis was 112 days. We also used Cox proportional models to calculate risk scores, checking for residual confounding. The distribution of overall risk scores for patients who received specialty care before their ulcer progressed to osteomyelitis or gangrene was slightly shifted towards higher risks of poor outcomes (S4 Fig).

Propensity scores improved the balance of important covariates between different specialty care groups (S2 Table). Inverse propensity score weighted (IPW) analysis yielded similar estimates for our primary and secondary outcomes in all strata (S3 Table and S5 and S6 Figs).

## Discussion

We report that receipt of specialty care is associated with improved outcomes for patients with diabetic foot ulcers, specifically longer times from ulcer diagnosis to disease progression, major amputation, or death. These benefits were observed regardless of the ulcer severity. However, over a quarter of our cohort did not receive specialty care, and, when provided, it typically was not multidisciplinary. Furthermore, the proportion of patients seen by each of the six specialties studied did not parallel the proportion of these specialties represented on multidisciplinary teams [7]. For instance, 82% of multidisciplinary teams included an endocrinologist, but only 10% of patients with a diabetic foot ulcer received care from an endocrinologist in this study. Nearly 75% of multidisciplinary teams included a vascular surgeon, yet only 10% of patients in this cohort were seen by a vascular surgeon. These discrepencies suggest the typical specialty care received by a patient with a diabetic foot ulcer differs significantly from that provided by multidisciplinary teams in academic, advanced limb salvage centers. Finally, specialty care that preceded, compared to followed, diagnosis of osteomyelitis or gangrene was associated with shorter times to disease progression, major amputation, or death.

We found a consistent trend towards improved limb salvage following specialty care, which is congruent with multiple systematic reviews of coordinated, multidisciplinary care [7,10,20]. However, when examining the single most utilized specialty in our dataset—podiatry—prior studies report mixed results. In the only randomized control trial, 91 patients with recently healed ulcers were randomized to free, monthly podiatric care versus standard care [21]. No

one underwent major amputation during the >1 year study, making it inadequately powered to observe a difference. Three retrospective cohort studies reported reductions in the odds of major amputation for patients receiving podiatric care, with most of this categorized as preventative care rather than our study's focus on ulcer treatment [22–24]. Greater geographic densities of infectious disease physicians, but not podiatrists, was associated with limb salvage [14,25]. Overall, our results confirm our hypothesis, corroborate prior retrospective cohort work, and substantiate expert opinion that specialty care is associated with improved outcomes for patients with diabetic foot ulcers [6,22–24].

Our finding that specialty care preceding, compared to following, diagnosis of osteomyelitis or gangrene was associated with shorter times to an event was counter-intuitive. It may indicate length bias, where event-free survival is overestimated due to the relative excess of cases that are slowly progressing for reasons unrelated to specialty care. These reasons might include less severity or complexity that cannot be elucidated with current claims algorithms. Alternatively, the fact that patients who received specialty care prior to ulcer progression lingered in the early stage strata much longer than their counterparts argues that specialty care might be contributing to delayed progression in particularly challenging cases. Prospective cohort studies that are able to provide a more granular assessment of ulcer severity and comorbidities would be needed to substantiate either possibility.

The association between specialty care and improved outcomes has important implications for policies, health systems, and individual clinicians. Policies that increase access to specialty care for patients with diabetic foot ulcers are likely to improve outcomes. Access to specialty care could be improved using multiple strategies including: 1) increased specialty care capacity, particularly in rural areas, 2) improved insurance coverage, and 3) interventions that reduce barriers to attendance. First, specialty care capacity could be improved through telemedicine, programs like Project ECHO that improve the reach of specialty care through primary care providers, periodic specialty care clinics staffed with traveling specialists, and permanent specialist placement in underserved communities [14,26–28]. Second, increasing the number of people with health insurance and broadening services covered by the plans would allow more patients to afford specialty care [29,30]. Expanded Medicaid coverage following the Affordable Care Act has been associated with reduced major amputations for patients identifying as other than non-Hispanic White, highlighting a policy-level intervention that may address healthcare disparities [30]. Third, patients report lack of transportation as a main barrier to timely medical care, and this was pressing for patients identifying as other than non-Hispanic White [31]. Travel vouchers and patient navigators, who could aide with community transportation resources in addition to other logistics, may help address this need [32].

Interventions at the level of health systems might also increase the proportion of patients seen by specialists and promote limb salvage. Within the Veterans Affairs healthcare system, increased coordination between services, such as primary and specialty care, was associated with decreases in amputations [33]. Similarly, multidisciplinary teams that prioritized triage into their services were particularly successful at limb salvage [7]. Conversely, the absence of primary care-specialty coordination was identified as a driving factor behind poor outcomes among rural patients with diabetic foot ulcers [34]. Rural patients who identified as Black were especially less likely to see a specialist, which may partially explain why these patients face some of the most daunting disparities in limb salvage [8,9]. Collectively, these studies suggest health systems that can improve coordination between services—particularly initial coordination between primary and specialty care—may improve outcomes and reduce disparities, especially rural disparities. The impact of such collaboration is echoed in the engineering literature

on multi-team systems, where coordination between teams influences outcomes more than coordination within teams [35–38].

Lastly, individual providers might positively impact outcomes by promoting timely consultation to specialty care. Clear *a priori* referral parameters might free time for primary care clinicians to address underlying factors such as glycemic control and smoking cessation [34]. Effective triage also hinges on the appropriate diagnosis of vascular disease, an area on which to improve for all clinicians managing diabetic foot ulcers [39,40]. Under-diagnosis of vascular disease likely precipitates missed opportunities for vascular surgery interventions and medical management of peripheral arterial disease, one of the greatest risk factors for poor outcomes in this population [41].

## Limitations

Like all studies, ours has limitations. Its retrospective cohort design precluded our ability to draw causal inference. However, a randomized controlled trial to determine causality would have questionable ethics; while the timing of specialty care is debatable, many would argue against a study arm that completely withheld it. Our analysis used a composite exposure of six different specialty care disciplines. While this enhances generalizability and allows for variations in practice based on what physiologic factor might be driving ulcer persistence (e.g., ischemia or infection), it limits our ability to comment on the association between improved outcomes and any single discipline. Studying the contributions of only six specialties also neglects important contributions by physicians and non-physicians who may be instrumental in caring for patients with diabetic foot ulcers in some settings. Most notably, these might include interventional cardiologists or radiologists performing lower extremity endovascular procedures and wound care nurses. We also did not explore the intensity or timing of specialty care, which would be important next steps for research. For instance, it would be useful to know what combinations of specialists offer the most effective care based on physiologic factors contributing to the ulcer. Podiatrists and infectious disease physicians might be a particularly good fit for ulcers complicated by osteomyelitis, while podiatrists and vascular surgeons might excel at caring for patients with gangrene. The timing of adding a second specialist could also be explored.

Because of our retrospective design, it is likely that our results are affected by survival bias, especially among those who received specialty care after entry into an ulcer stratum. Patients need to survive long enough to see a specialist. While this bias may contribute to a small, initial flattening of the Kaplan-Meier curves for those who receive specialty care and an initial steep decline for those who did not, we think its impact is minimal due to 1) the persistent association between specialty care and improved outcomes after controlling for comorbidities and ulcer severity, and 2) the improved outcomes among patients who received specialty care before diagnosis with an advanced stage ulcer, compared to those who received no specialty care. In this group, specialty care was initiated before ulcer progression and was not contingent upon surviving until they were able to receive specialty care after ulcer progression. Finally, more contemporary data would more accurately reflect current trends, particularly expanded use of telemedicine in the wake of the COVID pandemic.

## Conclusions

In summary, we observed that specialty care is associated with improved outcomes for patients with active diabetic foot ulcers, regardless of disease severity. Further studies that focus on the intensity and timing of specialty care would help elucidate the underlying mechanism and contribute towards the design of more effective interventions to increase specialty care at the

policy, healthcare system, and individual levels. Improving specialty care access might be a vital component to improving outcomes and advancing health equity among patients with diabetic foot ulcers.

## Supporting information

**S1 Fig. Histogram plots of surgical care timing relative to major amputation.**
(PDF)

**S2 Fig. Kaplan-Meier curves depicting the proportion of patients who have not undergone major amputation or died over the course of the study period for the early stage stratum (A), osteomyelitis stratum (B), and gangrene stratum (C).** Within each stratum, patients were characterized as having never received specialty care, having received specialty care before the diagnosis of osteomyelitis or gangrene (which applied to some of the patients who entered the cohort with a less severe ulcer and progressed to osteomyelitis or gangrene), after entering the stratum (i.e., being diagnosed with an early stage ulcer, osteomyelitis or gangrene, respectively).
(PDF)

**S3 Fig. Patients who received specialty care spent a longer time in less severe ulcer strata before their wounds progressed than patients who did not receive specialty care in earlier strata.** Analysis is based upon the most severe ulcer stage that a patient developed during the study period, either gangrene or osteomyelitis.
(PDF)

**S4 Fig. Density plots of risk scores for major amputation or death (left) or major amputation alone (right) among those with osteomyelitis (A) or gangrene (B).** Red corresponds to those who never received specialty care. Blue corresponds to those who received specialty care prior to ulcer progression. Yellow corresponds to patients who received specialty care after diagnosis of either osteomyelitis or gangrene.
(PDF)

**S5 Fig. Inverse propensity score weighted-adjusted Kaplan-Meier curves depicting event-free survival over the course of the study period for the early stage stratum (A), osteomyelitis stratum (B), and gangrene stratum (C).** Within each stratum, patients were characterized as having never received specialty care (red), having received specialty care before the diagnosis of osteomyelitis or gangrene (which applied to some of the patients who entered the cohort with a less severe ulcer and progressed to osteomyelitis or gangrene; blue), or after entering the stratum (yellow).
(PDF)

**S6 Fig. Inverse propensity score weighted-adjusted Kaplan-Meier curves depicting major amputation-free survival over the course of the study period for the early stage stratum (A), osteomyelitis stratum (B), and gangrene stratum (C).** Within each stratum, patients were characterized as having never received specialty care (red), having received specialty care before the diagnosis of osteomyelitis or gangrene (which applied to some of the patients who entered the cohort with a less severe ulcer and progressed to osteomyelitis or gangrene; blue), or after entering the stratum (yellow).
(PDF)

**S1 Table. International Statistical Classification of Disease and Related Health Problems, version 9, codes (ICD-9 codes) used to generate variables for the current study.**
(DOCX)

**S2 Table. Covariate balance before and after Propensity Score Weighting for each stratum.**
(DOCX)

**S3 Table. Inverse propensity score weighted and adjusted hazard ratios for the relationship between specialty care and the primary outcome of ulcer progression, major amputation or death, & the secondary outcome of major amputation or death, censoring for ulcer progression.**
(DOCX)

## Author Contributions

**Conceptualization:** Menggang Yu, Jennifer Mason Lobo, Justin J. Boutilier, Yao Liu, Meghan B. Brennan.

**Data curation:** Yingzhou Liu, Menggang Yu, Jamie N. LaMantia.

**Formal analysis:** Yingzhou Liu, Menggang Yu, Meghan B. Brennan.

**Funding acquisition:** Yao Liu, Meghan B. Brennan.

**Investigation:** Yingzhou Liu, Menggang Yu, Jamie N. LaMantia, Meghan B. Brennan.

**Methodology:** Yingzhou Liu, Menggang Yu, Jennifer Mason Lobo, Justin J. Boutilier, Yao Liu, Meghan B. Brennan.

**Project administration:** Jamie N. LaMantia.

**Supervision:** Menggang Yu, Meghan B. Brennan.

**Validation:** Menggang Yu, Meghan B. Brennan.

**Writing – original draft:** Yingzhou Liu, Menggang Yu, Jamie N. LaMantia, Meghan B. Brennan.

**Writing – review & editing:** Yingzhou Liu, Menggang Yu, Jamie N. LaMantia, Jennifer Mason Lobo, Justin J. Boutilier, Yao Liu, Meghan B. Brennan.

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
