## [Decision Letter · Decision Letter 0]

21 Aug 2023

PONE-D-23-21218Associations between specialty care and improved outcomes among patients with diabetic foot ulcersPLOS ONE

Dear Dr. Brennan,

Thank you for submitting your manuscript to PLOS ONE. After careful consideration, we feel that it has merit but does not fully meet PLOS ONE’s publication criteria as it currently stands. Therefore, we invite you to submit a revised version of the manuscript that addresses the points raised during the review process.

We look forward to receiving your revised manuscript.

Kind regards,

Tze-Woei Tan, M.D.

Academic Editor

PLOS ONE

Journal Requirements:

3. Please include a caption for figure 1.

Additional Editor Comments:

Please address the reviewers' questions and comments. Thank you.

Reviewers' comments:

Reviewer's Responses to Questions

**Comments to the Author**

1. Is the manuscript technically sound, and do the data support the conclusions?

Reviewer #1: Yes

Reviewer #2: Partly

2. Has the statistical analysis been performed appropriately and rigorously? 

Reviewer #1: Yes

Reviewer #2: No

3. Have the authors made all data underlying the findings in their manuscript fully available?

Reviewer #1: Yes

Reviewer #2: Yes

4. Is the manuscript presented in an intelligible fashion and written in standard English?

Reviewer #1: Yes

Reviewer #2: Yes

5. Review Comments to the Author

Reviewer #1: Congrats on your study. Thanks for your submission. A few comments:

- Methodology: for audience who is not familiar with your healthcare system, it may be useful to describe the typical journey/workflow for a patient with DFU. You may further contrast it with a patient who has vs has not been seen by a "specialist". There may be variations between Wisconsin and Illinois too

- Table 1: any value in further analysing patients who had been seen by more than 1 specialist?

- Table 2: it will be good to ascertain specific specialty care for each ulcer severity. For early stage, will be important to see podiatry/vasc/ortho. For OM, will be important to see podiatry/ID/ortho. For gangrene, will be important to see vascular/etc

- Thank you

Reviewer #2: The authors assess Medicare data to determine the impact of speciality care on limb salvage and outcomes in patient with PAD. The authors do a comprehensive analysis to achieve their results and prove their aim. However the manuscript lacks in the methodology. I have a few question:

1. The hypothesis needs to be reworded. There is no mention of organized or multi disciplinary speciality care.

2. Please add a clear aim for the manuscript

3. Please add a paragraph regarding information of Medicare database

4. Please include all the variable that were collected

5. Why was data assess till 2017. We are almost at end of 2023 and the data is about 5 years old

6. Please provide diagnosis code for diabetic ulcer

7. Also how did you define early stage, gangrene and osteomyelitis. Was this on diagnosis code? Please provide details

8. How was speciality information obtained? How did you choose the speciality?

What about wound care nursing?

9. Why did you choose endocrinology? Usually glucose is managed by pcp unless severe diabetes

10. Why orthopedics was included? Podiatry manages foot wound's

11 vascular surgery was a very low proportion of patients? Many places cardiologists or interventional radiologist are a part of vascular care.

12. Please clarify the stratum model as it unclear how this was assess.

May be add a flow chart

13. You talk about urban and rural as a dominant impact. May be add that in your aim or outcome measures

14. The follow up was very low? Was there longer follow up available?

15. Were patients with previous vascular intervention included? Do you have the data on that.

16. Please revise the discussion to highlight the significant points of the study

17. Please Ellaborate on the limitation section

18. Can reduce or merge some of the table and graph to avoid supplement

6. PLOS authors have the option to publish the peer review history of their article (what does this mean?). If published, this will include your full peer review and any attached files.

Reviewer #1: **Yes: **Zhiwen Joseph Lo

Reviewer #2: No

---

## [Author Response · Author response to Decision Letter 0]

17 Oct 2023

Reviewer #1’s Comments & Our Responses

1. Methodology: for audience who is not familiar with your healthcare system, it may be useful to describe the typical journey/workflow for a patient with DFU. You may further contrast it with a patient who has vs has not been seen by a "specialist". There may be variations between Wisconsin and Illinois too.

Author’s response: We now include a brief overview of current practice, which varies extensively, even within a state: “Currently, patients may be treated entirely by primary care providers or hospitalists. Others may be referred on to specialty care spanning a variety of different disciplines and varying considerably base, in part, on local workflows and regional availability” (page 4, lines 66-68). As you likely are aware, there is no typical journey, which adds to the importance of work like this.

2. Table 1: any value in further analyzing patients who had been seen by more than 1 specialist?

Author’s response: We think this is a very high yield area to explore in a subsequent paper. Data already exist to support a multidisciplinary approach, which is why it was not a direct focus of this article (current references 7, 10, and 20). However, we don’t know when adding a second specialist might be most useful. To answer this question, we need: 1) a larger dataset to increase our power because a fairly small proportion of our cohort was seen by >1 specialist, and 2) a dataset that contains dates of all specialty care visits, not just the first time a specialist saw the patient. This second piece of information would allow us to determine whether patients were seen over the same timespan by more than one specialist, as opposed to the first specialist ceasing to manage the patient after referring on to the second. Our current dataset does not allow this level of nuance. 

We hoped to convey some of these thoughts in our limitations section that initially read, “We also did not explore the intensity or timing of specialty care, which would be an important next step for research.” We now expand upon this by adding the following: “For instance, it would be useful to know what combinations of specialists offer the most effective care based on physiologic factors contributing to the ulcer. Podiatrists and infectious disease physicians might be a particularly good fit for ulcers complicated by osteomyelitis, while podiatrists and vascular surgeons might excel at caring for patients with gangrene. The timing of adding a second specialist could also be explored” (page 23, lines 390-395).

3. Table 2: it will be good to ascertain specific specialty care for each ulcer severity. For early stage, will be important to see podiatry/vasc/ortho. For OM, will be important to see podiatry/ID/ortho. For gangrene, will be important to see vascular/etc

Author’s response: We agree that these are great avenues for further research. Initially, we had hoped to investigate specialty combinations using this dataset. Unfortunately, the number of patients seen by multiple providers was too low in this regional dataset to glean meaningful insights. However, we included this as a key next step in our limitations section as above (page 23, lines 390-395).

Reviewer #2’s Comments & Our Responses

The authors assess Medicare data to determine the impact of speciality care on limb salvage and outcomes in patient with PAD. The authors do a comprehensive analysis to achieve their results and prove their aim. However the manuscript lacks in the methodology. I have a few question:

1. The hypothesis needs to be reworded. There is no mention of organized or multi disciplinary speciality care.

Author’s response: We studied specialty care in general, regardless of whether it was delivered as part of a multidisciplinary effort. We re-worded the hypothesis and added an aims statement to help clarify this: “We aimed to determine whether specialty care―regardless of whether it is part of an organized, multidisciplinary effort―is associated with longer times from ulcer diagnosis to progression, amputation, or death. We hypothesized that patients who received specialty care would have longer event-free survivals” (page 5, lines 77-81).

2. Please add a clear aim for the manuscript

Author’s response: Thank you for the suggestion. We now conclude the introduction with a clear aims statement followed by a hypothesis as detailed in the response above (page 5, lines 77-81).

3. Please add a paragraph regarding information of Medicare database

Author’s response: We added information about the Medicare database to the opening paragraph of the methods section, subsection termed data sources. In particular, we now specify that we used claims data from both inpatient and outpatient settings: “We evaluated a 100% cohort of Medicare fee-for-service beneficiaries residing in Wisconsin (2009–2017) and Illinois (2012–2017) using Part A and B claims data for ambulatory and hospitalized care, obtained through the Centers for Medicare and Medicaid Services Chronic Condition Data Warehouse” (page 5, lines 86-69).

4. Please include all the variables that were collected

Author’s response: Table 1 includes all variables that we collected. This is now stated in the methods section: “We collected three types of variables for this study, all of which are listed in Table 1: outcome variables, primary explanatory variables, and covariates” (page 7, lines 120-121).

5. Why was data assess till 2017. We are almost at the end of 2023 and the data is about 5 years old.

Author’s response: Unfortunately, the data we were able to purchase through Medicare only goes through 2017. We agree that a more contemporary dataset might be insightful. However, analyzing data generated during the height of the COVID pandemic also poses challenges as patients and providers were accessing and providing care atypically during this timeframe. We have added the following sentence to the limitations section: “Finally, more contemporary data would more accurately reflect current trends, particularly expanded use of telemedicine in the wake of the COVID pandemic” (page 24, lines 407-409). 

6. Please provide diagnosis code for diabetic ulcer

Author’s response: We now provide diagnostic codes for the different ulcer severities using Finke and colleagues’ validated algorithm as a new S1 Table first referenced on page 6, line 103.

7. Also how did you define early stage, gangrene and osteomyelitis. Was this on diagnosis code? Please provide details

Author’s response: We defined the early stage, osteomyelitis, and gangrene ulcer severity strata using Fincke and colleagues’ validated claims algorithm. We now reference the new S1 Table and the Fincke manuscript at the end of the following sentence to provide the reader with these details: “We constructed three strata based on ulcer severity (i.e., early stage, osteomyelitis, or gangrene) to account for confounding by indication, meaning that patients with more advanced ulcers would be more likely to both receive specialty care and have poor outcomes (Fig 1, S1 Table) [12]” (page 6, lines 108-111). 

8. How was speciality information obtained? How did you choose the speciality?

What about wound care nursing?

Author’s response: We identified the following specialists using Medicare taxonomy codes alone: orthopedic surgery, plastic surgery, and podiatry. We supplemented Medicare taxonomy codes with American Medical Association Masterfile data when identifying the following specialists: endocrinology, infectious disease, and vascular surgery. We supplemented because these providers often register with Medicare after finishing residency but before completing their specialty fellowships, leading to the Medicare taxonomy codes to miscategorize them as generalists. This information is included in the manuscript in two places within the methods: data sources (page 5, lines 89-93) and primary explanatory variable (page 7, lines 133-135). 

We chose these six disciplines because they were the six most frequent specialties represented within multidisciplinary diabetic foot ulcer teams, as identified in our systematic review of multidisciplinary teams (page 7, lines 130-133). This is not a comprehensive list, and it certainly excludes key nursing contributions, most notably wound care nurses. Claims data cannot reliably identify wound care nurses. We have now included this as a limitation: “Studying the contributions of only six specialties also neglects important contributions by physicians and non-physicians who may be instrumental in caring for patients with diabetic foot ulcers in some settings. Most notably, these might include interventional cardiologists or radiologists performing lower extremity endovascular procedures and wound care nurses” (page 23, lines 385-389). 

9. Why did you choose endocrinology? Usually glucose is managed by pcp unless severe diabetes

Author’s response: We included endocrinology because it was one of the six most common disciplines represented on multidisciplinary diabetic foot ulcer teams. It was actually the most common medical specialty represented, with 82% of teams involving an endocrinologist (Musuuza et al, reference 7). In our current study, just over 10% of all patients who developed a diabetic foot ulcer saw an endocrinologist. The lower proportion likely speaks to your point about PCPs spearheading glycemic control in the outpatient setting. 

We now highlight the discrepancy between the proportions of specialty care disciplines on multidisciplinary teams and the proportions of patients seen by these different disciplines in our study. We think this speaks to a very relevant care gap and thank you for bringing it to our attention. While the field is focused on multidisciplinary teams, the reality is that very few patients are cared for by them. The specialty care received by the majority of patients with diabetic foot ulcers is unlikely to be similar to that provided by these elite teams. To reflect this important point, we have added the following sentences to the first paragraph of the discussion: “Furthermore, the proportion of patients seen by each of the six specialties studied did not parallel the proportion of these specialties represented on multidisciplinary teams. For instance, 82% of multidisciplinary teams included an endocrinologist, but only 10% of patients with a diabetic foot ulcer received care from an endocrinologist in this study. Nearly 75% of multidisciplinary teams included a vascular surgeon, yet only 10% of patients in this cohort were seen by a vascular surgeon. These discrepancies suggest the typical specialty care received by a patient with a diabetic foot ulcer differs significantly from that provided by multidisciplinary teams in academic, advanced limb salvage centers” (pages 18-19, lines 286-294).

10. Why orthopedics was included? Podiatry manages foot wounds.

Author’s response: We included orthopedics because it was the second most common surgical specialty represented on multidisciplinary diabetic foot ulcer teams (Musuuza et al, reference 7). Two-thirds of teams had an orthopedic surgeon. Half had a podiatrist. In the current study, among patients who received specialty care, 20% of patients saw an orthopedic surgeon and nearly 60% saw a podiatrist. As mentioned above, the differences in specialty proportions are now highlighted in the discussion (pages 18-19, lines 286-294).

11. Vascular surgery was a very low proportion of patients? Many places cardiologists or interventional radiologist are a part of vascular care.

Author’s response: The proportion of patients seen by a vascular surgeon were certainly lower than the proportion of multidisciplinary teams that included a vascular surgeon, as highlighted in the discussion. You are correct that interventional cardiologists and radiologists are often performing endovascular interventions for these patients. In addition to drawing attention to the small proportion of patients seen by vascular surgery in the discussion (pages 18-19, lines 286-294), we also include exploring the role of interventional cardiologists and radiologists in the limitation section as follows: “Studying the contributions of only six specialties also neglects important contributions by physicians and non-physicians who may be instrumental in caring for patients with diabetic foot ulcers in some settings. Most notably, these might include interventional cardiologists or radiologists performing lower extremity endovascular procedures and wound care nurses” (page 23, lines 385-389).

12. Please clarify the stratum model as it unclear how this was assess.

Maybe add a flow chart.

Author’s response: Thank you for the idea of a flow chart to help clarify how patients could “advance” through the different ulcer severity strata. We built such a figure, current Figure 1, which is cited in both the methods (page 6, lines 111 and 117-118) and results (page 9, line 182).

13. You talk about urban and rural as a dominant impact. Maybe add that in your aim or outcome measures.

Author’s response: We think rurality is very important when discussing specialty care because rural patients have less access to specialty care. However, we did not include it as part of our aim in this paper because it was the focus of our prior publication (Taylor et al, current reference 9). In that paper, we found that only 29% of rural patients were seen by a specialist (compared to 32% nationally). However, there was no data to say that seeing a specialist was associated with improved limb salvage outside of multidisciplinary teams. This paper addresses that gap. Now we can say that rural patients: 1) have an increased risk of limb loss, 2) are less likely to see a specialist, and 3) specialty care, regardless of whether its delivered in a multidisciplinary setting, is associated with limb salvage. This logic suggests that improving access to specialty care for rural patients may help us close rural disparities in major amputation. We reworded part of the discussion to better emphasize this point: “Collectively, these studies suggest health systems that can improve coordination between services—particularly initial coordination between primary and specialty care—may improve outcomes and reduce disparities, especially rural disparities” (page 22, lines 360-363).

14. The follow up was very low? Was there longer follow up available?

Author’s response: Longer follow-up data is not available. However, we do not think that this was a significant limitation because 38% of our population either died or underwent a major amputation during the available follow-up, indicating we were studying a clinically relevant timeframe. This is congruent with other studies. In our national retrospective cohort study of Medicare patients with diabetic foot ulcers (2006–2011), 31% of the cohort either died or underwent major amputation within 2 years of developing a foot ulcer (PMID: 27993523). 

15. Were patients with previous vascular intervention included? Do you have the data on that?

Author’s response: We did not investigate prior vascular interventions, in part because we limited our cohort to patients whom we thought were likely to have incident diabetic foot ulcers. Specifically, they were required to have a two year baseline during which no ulcer was diagnosed. 

16. Please revise the discussion to highlight the significant points of the study.

Author’s response: We substantially reworked the beginning of the discussion to better highlight the significant points of our study. The initial paragraph now compares and contrasts the proportion of patients seen by different specialists and the proportion of those specialists on multidisciplinary teams (pages 18-19, lines 286-294). This juxtaposition highlights a gap between community and academic practice that is likely under-appreciated by those doing health services research on this topic. The second paragraph now concludes with a much stronger, summative statement: “Results confirm our hypothesis, corroborate prior retrospective cohort work, and substantiate expert opinion that specialty care is associated with improved outcomes for patients with diabetic foot ulcers” (page 20, lines 318-320). The prior third paragraph was entirely cut to help the reader focus on these take-home points before delving into the implications of this work at the levels of policy, healthcare systems, and individual providers. 

17. Please elaborate on the limitations section.

Author’s response: We now expand on the limitations section to include important next steps regarding a more nuanced study of specialty care, including evaluating care provided by different specialists, more than one specialists and specialty combinations that might be particularly effective for different ulcers. Specifically, the first paragraph of the limitations section now includes the following sentences regarding expanding the types of disciplines studied: “Studying the contributions of only six specialists also neglects important contributions by physicians and non-physicians who may be instrumental in caring for patients with diabetic foot ulcers in some settings. Most notably, these might include interventional cardiologists and radiologists performing lower extremity endovascular procedures and wound care nurses” (page 23, lines 385-389). The paragraph now concludes: “For instance, it would be useful to know what combinations of specialists offer the most effective care based on physiologic factors contributing to the ulcer. Podiatrists and infectious disease physicians might be a particularly good fit for ulcers complicated by osteomyelitis, while podiatrists and vascular surgeons might excel at caring for patients with gangrene. The timing of adding a second specialist could also be explored” (page 23, lines 390-395). We also added a sentence regarding the age of our dataset: “Finally, more contemporary data would more accurately reflect current trends, particularly expanded use of telemedicine in the wake of the COVID pandemic” (page 24, lines 407-409). 

18. Can reduce or merge some of the table and graph to avoid supplement

Author’s response: We strongly considered merging the Kaplan-Meier graphs for our primary (Fig 2, formerly Fig 1) and secondary outcome (S2 Fig) into a single figure or presenting both in the main text. However, we kept them separated because we think this improves clarity and focus of the main message.

---

## [Decision Letter · Decision Letter 1]

8 Nov 2023

Associations between specialty care and improved outcomes among patients with diabetic foot ulcers

PONE-D-23-21218R1

Dear Dr. Brennan,

We’re pleased to inform you that your manuscript has been judged scientifically suitable for publication and will be formally accepted for publication once it meets all outstanding technical requirements.

Kind regards,

Tze-Woei Tan, M.D.

Academic Editor

PLOS ONE

Additional Editor Comments (optional):

Thank you for addressing all the comments.

Reviewers' comments:

Reviewer's Responses to Questions

**Comments to the Author**

1. If the authors have adequately addressed your comments raised in a previous round of review and you feel that this manuscript is now acceptable for publication, you may indicate that here to bypass the “Comments to the Author” section, enter your conflict of interest statement in the “Confidential to Editor” section, and submit your "Accept" recommendation.

Reviewer #1: All comments have been addressed

Reviewer #2: All comments have been addressed

2. Is the manuscript technically sound, and do the data support the conclusions?

Reviewer #1: Yes

Reviewer #2: Yes

3. Has the statistical analysis been performed appropriately and rigorously? 

Reviewer #1: Yes

Reviewer #2: No

4. Have the authors made all data underlying the findings in their manuscript fully available?

Reviewer #1: Yes

Reviewer #2: No

5. Is the manuscript presented in an intelligible fashion and written in standard English?

Reviewer #1: Yes

Reviewer #2: Yes

6. Review Comments to the Author

Reviewer #1: Thank you for your revision. Looking forward to the subsequent follow-up papers, as described by the authors

Reviewer #2: The authors have addressed all the concerns to the best of their ability which have made the manuscript stronger. I would recommend to accept the manuscript

7. PLOS authors have the option to publish the peer review history of their article (what does this mean?). If published, this will include your full peer review and any attached files.

Reviewer #1: No

Reviewer #2: No

---

## [Editor Report · Acceptance letter]

5 Dec 2023

PONE-D-23-21218R1 

Associations between specialty care and improved outcomes among patients with diabetic foot ulcers 

Dear Dr. Brennan:

I'm pleased to inform you that your manuscript has been deemed suitable for publication in PLOS ONE. Congratulations! Your manuscript is now with our production department. 

Kind regards, 

on behalf of

Dr. Tze-Woei Tan 

Academic Editor

PLOS ONE